# Environmental and Economic Assessment of Repairable Carbon-Fiber-Reinforced Polymers in Circular Economy Perspective

**DOI:** 10.3390/ma15092986

**Published:** 2022-04-20

**Authors:** Elisabetta Abbate, Maryam Mirpourian, Carlo Brondi, Andrea Ballarino, Giacomo Copani

**Affiliations:** STIIMA-CNR—Institute of Intelligent Industrial Technologies and Systems for Advanced Manufacturing, National Research Council, Via Alfonso Corti 12, 20133 Milan, Italy; maryam.mirpourian@stiima.cnr.it (M.M.); carlo.brondi@stiima.cnr.it (C.B.); andrea.ballarino@stiima.cnr.it (A.B.); giacomo.copani@stiima.cnr.it (G.C.)

**Keywords:** carbon fiber, composites, automotive, ex ante LCA, economic assessment, LCC, recycling, repair, circular economy, sustainable manufacturing

## Abstract

The explosive growth of the global market for Carbon-Fiber-Reinforced Polymers (CFRP) and the lack of a closing loop strategy of composite waste have raised environmental concerns. Circular economy studies, including Life Cycle Assessment (LCA) and Life Cycle Costing (LCC), have investigated composite recycling and new bio-based materials to substitute both carbon fibers and matrices. However, few studies have addressed composite repair. Studies focused on bio-based composites coupled with recycling and repairing are also lacking. Within this framework, the paper aims at presenting opportunities and challenges of the new thermosetting composite developed at the laboratory including the criteria of repairing, recycling, and use of bio-based materials in industrial applications through an ex ante LCA coupled with LCC. Implementing the three criteria mentioned above would reduce the environmental impact from 50% to 86% compared to the baseline scenario with the highest benefits obtained by implementing the only repairing. LCC results indicate that manufacturing and repairing parts built from bio-based CFRP is economically sustainable. However, recycling can only be economically sustainable under a specific condition. Managerial strategies are proposed to mitigate the uncertainties of the recycling business. The findings of this study can provide valuable guidance on supporting decisions for companies making strategic plans.

## 1. Introduction

### 1.1. General Framework

Thermosetting Carbon-Fiber-Reinforced Polymers (CFRPs) are engineered composites made out of carbon fibers (CFs) as the reinforcement and epoxy resin as the polymer matrix [1], which acts as load transfer elements across fibers [2]. Among different types of resin (epoxy, phenolic, polyester, urethane and vinyl ester) [2], epoxy is the chosen resin when mechanical and resistance performance is required [3]. Indeed, CFRP composites are characterized by outstanding mechanical properties such as high stiffness, long life span, non-corrosive and high fatigue resistance [1,2,4]. Lightweight is an additional property of CFRP that can reduce, for instance, the overall weight of a vehicle up to 10% compared to steel and aluminum [5] with a reduction in fuel consumption or an increase in batteries duration for electric vehicles [6], resulting in a lower environmental impact during the use phase [7]. Indeed, CFRP composites are increasingly replacing other materials such as steel [8] and aluminum [9] in a wide range of sectors such as sports equipment, wind energy, aircraft, construction and automotive [4,10,11]. In recent decades, the global demand for CF increased from 16 kt to 72 kt [12], reaching 100 kt in 2019 [6]. Furthermore, the global demand for CF and CFRP is projected to reach, respectively, 117 kt and 194 kt in 2022, which portrays an annual compound growth rate, respectively, of 11.50% and 11.98% [13,14].

Although the undeniable potentialities of CFRPs, environmental and economic concerns on the spreading of CFRPs have been raised by many studies, including Life Cycle Assessment (LCA) and Life Cycle Costing (LCC) studies. LCA and LCC are the main adopted tools, respectively, for the environmental and cost burdens assessment in the entire life cycle of a product or a service [15,16]. Both tools allow not only the calculation of the environmental and cost impacts but also the identification of the life cycle stages with the most relevant impact and the potential drivers to reduce the impact [17]. Based on those studies, current challenges on CFRP composites are mainly associated with its waste production and management [8,18]. Firstly, albeit mechanical, thermal and chemical recycling are potential technologies for CF recovery [10] currently in the market [19], landfill and incineration are still the main routes of CFRP waste management. However, they are both in contrast with the Circular Economy concept [14,20,21]. Secondly, recycling of thermosetting CFRP is difficult because of its intrinsic properties, which do not allow it to be remolded and reshaped once it is cured [3,4]. Moreover, even when CFs are recovered, the recycled CFs could have a lower quality than virgin CFs [22]. Thirdly, 40% of CFRP scraps are generated during the manufacturing of the product [14]. Finally, several types of damages may occur both during product manufacturing, for instance, porosity or undesired bodies in the matrix, and during the use phase, for instance, delamination, matrix crack and fiber–matrix debonding [23]. However, in the case of a damaged CFRP product, repairing techniques are limited. Hence, its entire substitution is the most widespread approach, resulting in a further increase in costs and resources [24]. Weak CFRP waste management combined with the surging demand for CFRP has been leading to a tremendous amount of waste. It is estimated that the global CFRP waste will reach 20 kt per year by 2025. Furthermore, taking into consideration that the monetary value of CF is around 26.5 euro/kg [25], from an economic and environmental perspective, a considerable amount of composite waste accumulated every year results in a loss of valuable and energy intensive materials [12,14].

Among possible options for reducing the environmental and cost impact of CFRP composites, repairing, recycling and using alternative bio-based materials have been identified on legislative, industrial and research levels. Firstly, at the legislative level, Council Directive on End-of-Life (EoL) Vehicles defined the target on vehicles to reuse and recycle up to 85% by 2015, which is currently under review by the European commission [26]. At the industrial level, repairing technologies could extend the product lifetime, minimizing resource depletion [27] and costs [28]. If repair is not possible, a potential reduction in the environmental impact still occurs through used CF recovery [14]. Finally, at the research level, extensive R&D activities have been conducted in the material field to develop a new type of composite that can be recovered and reused [29] or to substitute CFs and traditional matrices with bio-based materials [7,30].

Within this framework, the environmental and economic assessment of composites in a circular economy perspective is urgent for the growth of CFRP, guiding potential future development and strategies to minimize its environmental burden. However, studies that focus simultaneously on the implementation of recycling, repairing and the use of bio-based materials are still missing. Section 1.2 provides details about the goal of the study. Section 1.3 and Section 1.4 provide the current status, issues and challenges, respectively, of LCAs and LCCs for the implementation of the above-mentioned actions on CFRP composites.

### 1.2. Goal of the Study

The goal of this paper is to perform an ex ante LCA coupled with LCC of new types of composites developed in a laboratory by Politecnico di Milano that can be repaired and recycled. In particular, this paper aims to:Highlight the importance of simultaneously evaluating repairing, recycling and using bio-based materials in CFRP sectors from a Circular Economy perspective to define the main drivers both on the environment and the economy;Couple LCA with LCC to obtain a broad assessment of the composites;Emphasize the importance of providing a preliminary environmental and economical assessment of emerging technologies;Outline the importance of needed efforts on repairing composite products to reduce their environmental impact, resource depletion and cost.

Section 2 describes the methodology adopted to perform both the LCA and LCC, including the scenarios developed, the data used and the assumptions made. Section 3 shows the results obtained regarding the environmental and economic aspects with a discussion of the main outcomes and limitations.

### 1.3. Life Cycle Assessment on Composites

Nowadays, the LCA tool is adopted by companies in order to estimate the current environmental footprint among several environmental categories and as a decision support tool to reduce their environmental impact [31]. As a matter of fact, the International Organization of Standardization (ISO) developed standards on LCA in order to consolidate the methodology [32] and provide guidelines to LCA practitioners and companies willing to apply this technique [33]. The tool estimates the current environmental impact of a product through its entire life cycle, from the raw material extraction to its End-of-Life (EoL), analyzing the entire supply chain. Moreover, it provides possible indications on reducing the environmental impact and a range of results according to the developed scenarios. LCA is divided into four iterative steps, which are *Goal and scope definition*, *Life Cycle Inventory (LCI)*, *Life Cycle Impact Assessment (LCIA)* and *Interpretation* [33]. Regarding the LCI step, primary data are directly provided by companies; secondary data are available from datasets and literature [34].

Although the LCA tool is mainly applied to existing technologies [31,35,36], a preliminary environmental assessment of emerging technology further enhances the knowledge of this technology from an environmental point of view [37], improving its design and further developing the product on an industrial scale [38]. As a consequence, this could attempt at obtaining a product in the market with as little environmental impact as possible [35]. This is the case of the ex ante LCA. It simulates the environmental impact of an emerging technology on an industrial level, for instance, a technology developed at laboratory, in order to (1) estimate its environmental impact among possible scenarios in case it will be developed on an industrial scale [31,39] and (2) compare it with existing technologies that provide the same or similar service [40]. Difficulties and challenges have emerged in developing an ex ante LCA mainly regarding the data collection [41] and scenario selection due to the absence of data, differences between industrial and laboratory scale and uncertainties on future scenarios of development [42] and large-scale technology diffusion on the market [43]. Indeed, the ex ante methodology is applied in order to solve the following two main issues related to primary data from a laboratory: (1) scale-up from consumption to industrial scale [42,44] and (2) unavailable primary data [42,45]. On one hand, technologies used as well as consumption of energy and materials on a laboratory scale are different from the ones at the industrial level. As a matter of fact, the consumption of raw materials and energy can be much higher on a laboratory scale than the one on the industrial scale [42,46]. On the other hand, data on energy and auxiliary materials consumption are usually not available at the laboratory scale [42], and hence, calculations are needed in order to compute the LCI. Efforts on establishing a methodology through which an ex ante LCA can be performed has been made by different authors [40]. Cucurachi et al. [31] identified five types of ex ante LCAs in order to uniform the present literature on this topic, provided a definition of ’emerging technology’ and outlined the main challenges of performing ex ante LCAs. Moni et al. [42] focused on the challenges to perform LCA of emerging technologies, identifying possible pathways and recommendations in order to overcome those limits. Piccino et al. [46] provided an operational methodology of an ex ante LCA through which data on a laboratory scale can be transformed to industrial-scale data. However, few LCAs directly applied ex ante methodology in real case studies.

Concerning the application of LCA to composites, many studies can be found in the literature. LCAs on glass fiber polymers [47,48] and carbon fiber polymers [49,50,51] as well as thermoplastic [52,53] and thermosetting composites [5,52] among different applications, for instance, automotive [8,49,54], aviation [7,55] and construction [48,56,57] sectors. More recent comparative LCAs on CFRP waste management included mechanical, chemical and thermal recycling as alternatives to landfills and incineration [58]. Meng et al. [12] analyzed recycling technologies of CF, concluding that all recycling routes could achieve environmental benefits in terms of Global Warming Potential (GWP) and Primary Energy Demand.

As concerns composite repair, the use of composite patches is one the most suitable methods to repair damages to composite products [59]. In particular, Mohammadi et al. [59] discussed the mechanical and structural advantages of the use of composite patches. However, LCA studies on this topic are still missing. Tapper et al. [60] mentioned maintenance and repair of vehicles which were, however, excluded in the study. Raugei et al. [49] analyzed a thermoplastic composite outlining its properties that enable repairing and recycling. The LCA also analyzed the maintenance stage of a vehicle, which included a percentage of substitution due to damages in its entire lifetime. However, the repairing was not included in the LCA analysis. As concerns the ecodesign efforts for material circularity, efforts have been made both on substituting the CF and traditional matrices with bio-based materials. Hermansson et al. [30] outlined the environmental benefit of using lignin as feedstock to produce CF, collecting several LCA studies compared to the traditional CF. Cotton, flax, hemp, jute and bamboo are some of the bio-based fibers that can substitute the traditional PAN used as a precursor to the CF [7,48]. Moreover, *Mischantus* fiber was analyzed by Roy et al. [61] as potential promising feedstock in substitution of the traditional CF. However, incineration with energy recovery of this type of material was mentioned as a good alternative to chemical and mechanical recycling, which are instead less feasible and efficient [7]. As concerns bio-based resin, the use of wood pulp or biofuels could contribute to reducing the environmental impacts of composites [7]. At present, LCA studies on composites that simultaneously focused on repairing, recycling and using bio-based materials are missing. Furthermore, ex ante LCAs that preliminarily evaluate the potential of new composites are lacking. Within this framework, this LCA attempts to provide (i) preliminary potentialities of the new composites for further development on an industrial scale and (ii) an environmental assessment of the implementation of recycling, repairing and using bio-based materials from a circular economy perspective. The preliminary literature review on composites performed in this study is reported in Appendix A.

### 1.4. Economic Sustainability of Composites EoL

The economic assessment of a new project or a technology under different system configurations is of paramount importance to attain competitive edge since it helps business management to compare the profitability of the new business under a wide range of scenarios. To increase the competitive advantage of a product, the Life cycle cost (LCC) analysis is a proper tool that is used to compare the profitability of the new business under a broad range of hypotheses [16,62,63]. The U.S. military first introduced the LCC concept in the 1960s [64]. The idea was then used by different industries, such as energy, transportation, healthcare, etc., to assess a project’s economic sustainability and helped investors compare the cost-effectiveness of alternative business decisions [16]. LCC analysis starts with the cost breakdown structure (CBS) and revenue breakdown structure (RBS) of a product or a new project. This approach measures and calculates the costs and revenues of the product at its different stages of life. Then costs and revenues are discounted using the discount rate. The discount rate represents the rate of inflation at the time when the financial model is performed [64]. From a higher-level standpoint, the LCC procedure aggregates all the discounted cash flows incurred over a project’s entire life span. These discounted cash flows can be used to identify the profitability of the new project based on financial indicators such as net present value (NPV) and discounted payback period [65]. Although economic assessment and LCC are widely applied in different sectors, a very limited number of studies have investigated the economic sustainability of composites EoL solutions in the literature. Many studies on composites have evaluated the economic sustainability of products during the manufacturing and use phase. For example, Witik et al. [58] investigated the financial sustainability of CF composite products using diverse manufacturing processes (autoclave vs. out-of-autoclave). Delogu et al. [25] carried out a comparative analysis to evaluate the economic performance of CFRP during the manufacturing process compared to the use phase. Strogonov [66] analyzed the cost reduction of products made out of composites compared to steel products during the manufacturing process. However, very few studies have focused on evaluating the economic performance for the EoL of composite waste. In this regard, for example, Castella et al. [67] assessed the economic sustainability of a composite product in the automotive industry during different value chain stages: raw material fabrication, manufacturing, use phase and the end of life. It was found that using hybrid composites instead of steel can lower the lifecycle cost by 16%. La Rosa et al. [62] also focused on the economic performance of carbon fiber (CF)-thermoset composite during manufacturing and solvolysis recycling. The finding of this study proved that the chemical recycling is not economically sustainable. However, research on this topic is still lacking. In a broader sense, it can be argued that most studies on composite EoL were focused on the technical performance of recycling technologies [1,4,10,68]. Among these studies concentrating on the closing loop techniques, a few have briefly evaluated the economic impacts of recycling technologies. For example, Karuppannan Gopalraj and Kärki [10] carried out an extensive literature review on the technical aspects of the mechanical recycling, thermal recycling and chemical recycling and briefly pointed out that thermal and chemical recycling can be economically sustainable under certain conditions. Finally, it can be stated that although within the circular economy paradigm, repair is defined as one of the most effective approaches to address waste [69] and composite repair has been proven technically feasible [59], the economic assessment on this topic is still missing. Therefore, this paper aims to address this knowledge gap by evaluating the economic sustainability of EoL composites through repair and thermal recycling (CO${}_{2}$ assisted pyrolysis).

## 2. Materials and Methods

This section describes the methods adopted to perform the LCA and LCC of a couple paddle of shifters used in the automotive sector. In particular, Section 2.1 describes the methodology adopted for the LCA, which includes the goal of the study, the functional unit, the system boundaries, the scenarios developed, the data collection of the LCA study, the characterization method used and impact categories. Finally, Section 2.2 focuses on the collection of data used for the costing analysis.

### 2.1. Environmental Assessment

#### 2.1.1. Goal and Scope Definition

This ex ante LCA is used as an exploratory methodology for the introduction of new technologies on the industrial level. The scope of this LCA is to provide a comparative analysis of CFRPs, and scenarios were built according to the following criteria:Type of materials used for the production of the CFRP product;Possibility of repairing the damaged part of the CFRP product;Different End-of-Life (EoL) of the CFRP.

As concerns the first criteria, two types of composites were developed. Generally, the composite is made of CFs, an epoxy resin [70] and a curing agent that define the properties of the composite [71]. In this study, both composites were developed using the same type of epoxy resin and two different types of dynamic curing agents. The first type of composite was produced using a fossil-fuel-based dynamic curing agent (i.e., CFRP-ff); the second type of composite was produced using a bio-based dynamic curing agent (i.e., CFRP-bio).

As concerns the second criteria, this LCA compares two scenarios of each aforementioned composite in order to evaluate the potential environmental benefits of repair during the use phase. On one hand, the impossibility of repairing the composite requires the substitution of the product. This means that the waste management of the damaged CFRP product and the production of the second CFRP product are considered. On the other hand, the possibility of repairing implies that no additional CFRP product is required to be manufactured. In this case, only the repairing process was included. In this study, one damage in the entire life of the car was assumed. The third criteria aimed at comparing different EoLs. In accordance with other studies on composites [8,12,49,52], as the base scenario of LCA, 50% of the CFRP waste was assumed to be sent to a landfill, and the remaining 50% was assumed to be sent to incineration. Two additional scenarios, respectively, mechanical and thermal recycling, were analyzed to further evaluate the environmental benefits of both recycling compared to the current waste management procedure. Further details on scenarios are reported in Section ‘Scenario Description’.

Based on the properties of the developed CFRP, it can be used in automotive and aeronautic sectors. The functional unit (FU) is one couple of paddle shifters used in the automotive sector. The system boundaries are shown in Figure 1. The chosen LCA database is ecoinvent attributional, cut-off v3.6. Additional data used for the inventory were taken from the literature. The chosen LCIA method is ReCiPe in accordance with other LCA studies on composites [53,72,73,74].

##### Scenario Description

Six scenarios, listed in Table 1, were defined according to the criteria described in Section 2.1.1. Figure 1 describes the steps included in the study for each scenario. Groups of scenario A and B represent the life cycle, respectively, of CFRP-ff and CFRP-bio. Scenario A1 and B1 are representative of the most likely situation of a composite EoL: landfill and/or incineration [19]. Moreover, in the case of damage during the use phase, the composite is substituted with a new composite, which results in another production and manufacturing process of the composite.

Scenario A2 and B2 include the repairing of the composite after the damage instead of its substitution. At the end of the entire life of the car, the composite is sent to mechanical recycling.

Scenario A3 and B3 include the repairing of the composite after the damage instead of its substitution. At the end of the entire life of the car, the composite is sent to thermal recycling. Both thermal and mechanical recycling allow the recovery of epoxy resin and CF.

#### 2.1.2. Life Cycle Inventory

##### Production of the Prepreg

The production of prepreg, which is the pre-impregrated CF with the epoxy resin, is the most widespread manufacturing route at the industrial level [14]. Firstly, the epoxy resin is mixed with a curing agent, and secondly, the CFs are pre-impregnated with the cured epoxy resin in order to obtain the prepreg [55]. The production of CFRP-bio requires a further transesterification catalyst to be added to the mixing and additional pretreatment for the activation of the curing agent prior to the mixing of the epoxy resin with the curing agent. Figure 2 summarizes the production processes of prepreg at the laboratory scale for both composites.

An ex ante evaluation was needed in order to estimate the energy and materials consumption at the industrial level starting from the data at laboratory scale. Table 2 shows the main differences between the primary data provided from laboratory tests and the primary data needed for the LCA modeling usually provided at the industrial scale. The transformation of primary data from laboratory to industrial scale was performed for the following prepreg production steps shown in Figure 2: *Mixing of curing agent and epoxy resin*, *mixing of curing agent, catalyst and epoxy resin* and *pretreatment of curing agent and catalyst*. The scale-up of those steps was conducted according to the methodology introduced by Piccinno et al. [46]. In particular, for each process listed in the first column of Table 2, the equations provided by Piccinno et al. [46] were implemented, and assumptions were introduced in order to obtain the inputs to the LCA model starting the primary data provided from laboratory tests (i.e., second column of Table 2). The overall electricity consumption of the first step of prepreg production at the industrial scale was estimated with the equations shown in Table 3 taken from Piccinno et al. [46]. The ratio between the epoxy resin and the curing agent was taken from De Luzuriaga et al. [3]. Adjustments were made for CFRP-bio to include the catalyst in the prepreg composition.

Table 4 and Table 5 summarize, respectively, the two steps of production, outlining the main information on materials and energy consumption both for CFRP-ff and CFRP-bio. The electricity consumption for the impregnation of the CF in Table 5 was taken from Song et al. [54]. Background data such as epoxy resin, CF, the fossil-fuel-based curing agent, the bio-based curing agent and the catalyst were taken from the literature and ecoinvent database. The epoxy resin was modeled according to ecoinvent database *market for epoxy resin, liquid|epoxy resin, liquid|Cutoff, U-RER*. CF production was modeled according to literature data. Data from Khalil [55] were used to model CF production from PAN. Data from Duflou et al. [8] were used to model Polyacrylonitrile (PAN) starting from Acrylonitrile, which is modeled in the used database. The fossil-fuel-based curing agent was modeled according to the ecoinvent database with an aromatic amine *market for meta-phenylene diamine|meta-phenylene diamine|Cutoff*. The material was chosen in accordance with studies on epoxy resins [75,76]. The bio-based curing agent derives from different routes, e.g., from fructose, polysaccharides [77], starch, glucose [78] and lignin [79]. Lignin biomass was used as reference production routes starting from tetrahydrofuran (THF), and data from Kim et al. [79] were used. The catalyst was modeled according to the ecoinvent database with the *market for ammonium thiocyanate|ammonium thiocyanate|Cutoff* process.

##### Manufacturing the Product

Manufacturing the CFRP product mainly includes the cutting phase of the prepreg and the lay-up phase. The latter phase is the core manufacturing process through which the prepreg is modeled in order to obtain the final shape of the desired product, which, in our case, is on a couple of paddle shifters. The lay-up phase can be performed with different technologies [72]. Although primary data from a laboratory were available, the manufacturing of the paddle shifters was modeled according to literature data for two main reasons: i. experiments in a laboratory are performed with small quantities of materials and a higher level of auxiliary materials consumption and scraps and waste generation compared to the expected quantities consumed and generated at the industrial level [42,46]; ii. the autoclave is one well-established technology at the industrial level for CFRP products. Hence, the auxiliary material and energy consumption were reasonably assumed not to be linked to the prepreg composition.

The manufacturing phase was modeled according to Forcellese et al. [72], including the mold and the master production. The data inventory is reported in Table 6. The auxiliary materials used during the lay-up phase, such as the vacuum bag in Polyamide 66, the breather (Polyethylene terephthalate), the release film (Polytetrafluoroethylene) and the release agent, were assumed to be sent to a landfill at the end of the production since they were not reusable. According to the scope of the study, the FU is the production of a CFRP product in the automotive sector. As a consequence, the manufacturing of the product is followed by the final assembly to the car. However, according to Raugei et al. [49], the environmental impact due to vehicle manufacturing is low compared to the other life cycle stage. Hence, assembly was assumed negligible compared to the entire life cycle, and it was not considered in the study. Further details about distances and transportation are described in Section ‘Transportation’.

##### Use and Repairing

The use phase of the paddle shifters is directly linked to the use phase of the car, which is mainly related to fuel consumption determined by, for instance, the shape and weight of the car [5]. The latter depends on the materials used. As a matter of fact, the use phase of a car component is usually considered in LCA studies in order to evaluate its indirect environmental impact [73]. However, the weights of the two CFRPs analyzed in this study are almost the same. Moreover, the weight of the FU is 76 g, which is reasonably low compared to the weight of the whole car, which is usually in the range of 500–1000 kg [80]. Hence, the comparison of their environmental impact during the use phase was reasonably assumed not to be relevant to the comparative LCA.

Assuming the overall life cycle of the paddle shifters equals the life cycle of the car, damage to the paddle shifters in the entire life cycle of the car was considered in the study. Two main options were analyzed in this analysis. The first option, represented by scenarios A1 and B1, describes the substitution of the product with a new one. As shown in Figure 1, the damaged product is sent to a landfill or for incineration, and a new product is manufactured with the same technologies of the first composite. It was assumed that both paddle shifters were damaged and substituted. The second option represented by scenarios A2, A3, B2 and B3 describes repairing of the damaged part and avoiding the substitution of a new paddle shifter. The repairing of the damaged part is performed using laser technology under development at the laboratory scale. Consequently, the operation of substitution with patches is not required. Based on the development of this technology, it was assumed in the present analysis that 1 cm${}^{2}$ of resin was damaged. Although the electricity is the main contribution to this phase, the value is omitted for confidentiality reasons. No waste is generated during the process.

##### End of Life

Scenario A1 and B1 represent the most likely situation and the worst scenarios from a Circular Economy prospective [10,19]. As a matter of fact, the paddle shifters are sent to a landfill and for incineration at the end of their life. Moreover, as described in Section ‘Use and Repairing’, after one paddle shifter is damaged, it is sent to a landfill and/or for incineration, and a new one is manufactured. The process of landfill and incineration were modeled using the ecoinvent database.

Scenario A2 and B2 describe the situation in which the paddle shifters are mechanically recycled. The SELFRAG is a technology for mechanical recycling that aims at disintegrating the composite through the principle of electrodynamic fragmentation: the composite is subjected to electrical discharges that break the composite. The process occurs using water [81,82]. SELFRAG technology has been used in laboratories in order to understand the potentiality of recycling both the composite and the resin. Currently, 60% of the CF can be recycled, while the possibility to recycle the resin is still uncertain and under study. Hence, an overall 60% of recycling was considered in this study without specifying the composition of the recyclates. Data were collected from the official report of SELFRAG [81]. Inventory data are reported in Table 7.

Scenario A3 and B3 analyze the thermal recycling through CO${}_{2}$-assisted pyrolysis. Thermal recycling of composite operates at high temperatures in the range of 350–750 °C [10,18,19,83], and at the end of the thermal process, the CFs are recovered. Korec is CO${}_{2}$-assisted pyrolisis technology commercially available to recover glass fiber composites; it is currently being tested in laboratories in order to analyze the potentiality of CFRP recycling. Inventory data are reported in Table 8. According to KOREC data, the glass fiber is recovered with an efficiency of 99% [84]. The same recovery efficiency was assumed for the CF. Eighty-five percent of the resin can be recovered, and up to sixty percent of recycled resin can be mixed with virgin resin [85]. CO${}_{2}$ is recovered and reused internally in the process, and consequently, its consumption was assumed to be negligible. The electricity consumption of the process was taken from Nunes et al. [52], with slight differences in the process. During the process, solid residue is generally generated in the case of inert mineral fillers in the composite. Non-condensable gas is produced during the process and is used for energy purposes, fulfilling the energy requirement. However, it was not possible to characterize and quantify the solid residue for our specific composite due to its small quantities.

##### Transportation

Table 9 shows the transportation included in the study from the raw material production until the composite’s EoL. Since the analysis was preliminary, and it was not focused on evaluating the environmental impact of a product that is available on the market, the supply chain was unknown. Hence, the following assumptions were introduced. The production of the prepreg and the manufacturing of the paddle shifters were assumed to be located in Italy in different cities. The production of the CF, the curing agents and the epoxy resin was assumed to be located in Europe by estimating an average distance to the production site of the prepreg. Distance from the manufacturing site to the car assembly site was not considered since the assembly phase was excluded from the system boundaries. The repairing site was assumed to be the same as the manufacturing site. Distance from the user to the repairing site was conservatively assumed to be 500 km due to the high variability of the car user’s location. Information on mechanical and thermal recycling locations were estimated from information on companies provided by Giorgini et al. [19]. Landfill and incineration were assumed to be located 50 km from the collection point.

#### 2.1.3. Life Cycle Impact Assessment Method

The LCIA was performed using OpenLCA software, version 1.11.0 (GreenDelta, Berlin, Germany) with the Ecoinvent 3.6 database for the LCI phase and ReCiPe mid-point (hierarchist) as LCIA methodology. The chosen LCIA methodoloy allows the assignment of the LCI results to indicator results according to the chosen environmental impact categories, which represent the product profile [33]. The following eighteen mid-point metrics were adopted to perform the LCA. Fine particular matter formation (kgPM${}_{2.5}$eq.), Fossil resource scarcity (kgoileq.), freshwater ecotoxicity (kg1,4-DBeq.), freshwater eutrophication (kgPeq.), Global warming (kgCO${}_{2}$eq), human carcinogenic toxicity (kg1,4-DCBeq.), human non-carcinogenic toxicity (kg1,4-DCBeq.), ionizing radiation (kBqCo-60 eq), land use (m${}^{2}$a crop eq.), terrestrial acidification (kgSO${}_{2}$eq.), marine ecotoxicity (kg1,4-DCBeq.), marine eutrophication (kgNeq.), Mineral resource scarcity (kgCueq.), Ozone formation Human health (kgNO${}_{x}$ eq), Ozone formation terrestrial ecosystems (kgNO${}_{x}$ eq), terrestrial acidification (kgSO${}_{2}$eq), terrestrial ecotoxicity (kg1,4-DCBeq.), Stratospheric Ozone depletion (kgCFC11eq.) and water consumption (m${}^{3}$) were quantified for each stage of the paddle shifters’ life cycle.

### 2.2. Economic Sustainability Assessment

To assess the economic sustainability of the new EoL composite solutions, which are repair and thermal recycling, multiple supply chain perspectives have been taken into account: the one of the composite parts producers and the one of composite materials recyclers. The economic assessment of producer is performed for the two new composites developed in a laboratory, i.e., CFRP-ff and CFRP-bio, as described in Section 2.1. Regarding the repair, producers of composite parts are supposed to integrate the new CF composites into their production system. Due to the properties of the new composites, they can be repaired [86]. From a business standpoint, the producer would use the new material only if it could take advantage of the repair of the material. Since the new CF composite price is higher than the conventional ones, for the manufacturer, the main incentive of using the new materials is to exploit the repair opportunity to address the defective products coming from the production process. Therefore, in this new paradigm, a producer would also become a remanufacturer by integrating the repair facilities into its existing production plant. Here, it is important to highlight that, based on the interviews with the production manager of the case study company, it becomes evident that one of the main issues of CF composite products is their high defect rate. Thus, in our hypothesis, the potential increase in the cost of the raw materials could be compensated by the potential savings due to the introduction of in-house repair practices enabled by the new composites. In addition, another aspect of this business is the fact that product repair helps the producer abolish its landfilling cost. In the current context, when products made from conventional CFRP have a defect during the production process, they cannot be repaired and are sent to a landfill. Thus, the manufacturer bears their landfilling cost, while in this new setting, producer landfilling costs associated with defective parts are eliminated. Regarding the system boundaries, the economic assessment consists of material production, manufacturing, and repair of defective products (products that are damaged during the production process) for a producer of the paddle shifter.

On the contrary, the economic assessment of thermal recycling represents an entirely new business for a composites recycler. In this framework, the recycler receives CFRP waste that is recyclable. The new thermal recycling technology enables both carbon fibers (CF) and resin to be recovered and sold for secondary applications. In this study, we restricted the economic simulation to the sales of CF, neglecting the possible revenue stream from recycled resin. This was justified by the relatively higher added-value of the fibers compared to resin and to the multiple criteria that have to be assessed for recycled resin to be used in secondary applications, which cannot be forecast at the moment due to the novelty of the developed material. The system boundaries consist of a recycler’s economic sustainability receiving recyclable composite waste, i.e., CFRP-ff and CFRP-bio wastes, and selling the recovered CF. The economic sustainability of composite parts manufacturers and composites recyclers has been assessed through LCC techniques. In this study, parametric modeling is used to perform the LCC analysis. Parametric models provide a high level of flexibility in changing the parameters. These models are appropriate for identifying cost drivers and carrying out the sensitivity analysis. They allow the user to define different scenarios under a wide range of assumptions.

#### 2.2.1. Case Study Definition

The case studies considered in the economic assessment of a producer of CFRP parts and the recycler are presented in this section.

Two main industrial cases are taken into account for the repair producer. One case investigates the economic sustainability of the repair business when the paddle shifter is made from CFRP-ff, and the other assesses the profitability when the product is made from CFRP-bio.One main industrial case is taken into account for the thermal recyclers receiving either CFRP-ff or CFRP-bio waste. From an economic standpoint, the recycling processes of CFRP-ff waste or CFRP-bio waste are very much alike for the recycler for the following three reasons. First, the recycling process of both materials is precisely the same. Second, the acquisition cost of CFRP-ff and CFRP-bio wastes is the same for a recycler. Based on the interview with the operation manager of a recycling company, it is hypothesized that the recycler would pay no waste acquisition cost since companies in charge of dismantling would be relieved from sustaining recycling costs. In this framework, the recycler only bears the transportation cost of waste to its plant. Finally, since the percentage of CF in new CFRPs materials is almost the same (49% for CFRP-ff and 48% for CFRP-bio), it becomes evident that the financial results of a recycler receiving CFRP-ff waste or CFRP-bio are almost the same. In addition, in this study, the economic modeling of the recycler is not only restricted to the paddle shifter. Since recycling is a high-volume business in which its sustainability is based on the economy of scales, and the baseline product considered in our case study is a low-volume niche component produced with a limited quantity of raw materials. Thus, we hypothesized that the recycler receives various products from multiple customers built with the new recyclable material in order to aggregate massive volumes to be treated.

#### 2.2.2. Cost and Revenue Breakdown Structure of Manufacturer

Based on the various steps of the production and repair processes discussed in Section 1.4, specific CBS and RBS have been defined for the producer. The input data related to the manufacturing costs of the paddle shifter were collected during two rounds of interviews in 2021, with the production manager, technical manager and marketing manager of the case study company, as shown in Table 10 and Table 11. In addition, data related to laser technology were provided by Politecnico di Milano in 2021. The costs of the business model of a producer that integrates new CFRP composites into its production process and repairs its defective parts are as follows:The manufacturing cost includes the machinery, depreciation, labor, energy and overhead cost. Due to data confidentiality, the manufacturing cost of product is shown as a percentage of the whole product cost;The material cost of a product represents the cost of new repairable/recyclable composites: CFRP-ff or CFRP-bio. Due to data confidentiality, the material cost of product is shown as a percentage of the whole product cost;The investment cost of repair facilities includes purchasing the laser technology and its accessories;The maintenance cost includes one operator cleaning and taking care of the optics of laser technology;The repair cost consists of the energy consumption and labor needed to repair the damaged area. As described in Section ‘Use and Repairing’, the dimension of the damaged area is considered to be 1 cm${}^{2}$. Given that the energy consumption of 1 cm${}^{2}$ is 0.111 kWh, and as the time needed to heal the damaged part is 5 s, the total repair cost of 1 cm${}^{2}$ is EUR 0.045 (as shown in Equations (Equation 1)–(Equation 3)).
(1)$$Total\;cost_{repair}=Energy\;Cost_{repair}+Labor\;Cost_{repair}$$
(2)$$Energy\;Cost_{repair}=Electricity\;consumption*Cost_{Electricity\;power}$$
(3)$$Labor\;Cost_{repair}=Time\;needed\;to\;repair\;the\;damaged\;area*Cost_{Personnel\;wages}$$

The revenues stream of manufacturer includes:Selling of the product made either from CFRP-ff or CFRP-bio. Here, it is important to mention that the selling price of the paddle shifter made from the new repairable/recyclable CFRP is assumed to be the same as the product made from conventional CFRP. Since, based on the interviews with the case study company, the market might not accept an increase in the product price, and the best strategy is to keep the selling price at the same level of the conventional product. Here, it is also important to mention that the profit margin of this product is relatively high (50%) due to the high value-added of composite in the high-end automotive market.

#### 2.2.3. Cost and Revenue Breakdown Structure of Recycler

Based on the various steps of the recycling processes discussed in Section 2.2, specific CBS and RBS have been defined for the recycler. The input data of the thermal recycling business model, as shown in Table 12, were provided by Politecnico di Milano, an operation manager of a thermal recycling company located in Northern Italy and KOREC technology in 2021.

In the following, all the recycler’s costs and then the recycler’s revenue stream is presented:The investment cost includes purchasing the CO${}_{2}$-assisted pyrolysis facilities. The investment cost of thermal recycling is EUR 550,000 for a plant of 250 ton/year;The maintenance cost includes the price of 1 h of work of an operator who cleans, lubricates and adjusts the thermal recycling machine after every 8 h cycle;The recycling cost includes the processing cost in terms of the depreciation, labor and energy cost;The overhead cost includes all administrative and accounting fees;The transportation cost includes delivering the EoL of CFRP waste to thermal recycler facilities. As mentioned in Table 9, the distance between the waste management company and the recycler is assumed to be 1000 km. Within these assumptions, the operation manager of the recycling company provided us the transportation cost of waste to be 0.20 EUR/kg, with a coefficient of +0.05 EUR/kg and −0.05 EUR/kg.

The revenue stream of the recycler is:Selling of the recycled CF to secondary applications. As mentioned in Table 8, the recovery rate of CF within the CO${}_{2}$-assisted pyrolysis is 99%. Here, it is important to mention that the market price for selling the recycled CF is 5 EUR/kg, with a coefficient of +1 EUR/kg and −1 EUR/kg.

#### 2.2.4. Assumptions

The assumption that were made to carry out the economic assessment are as following:The cost estimation of new repairable/recyclable composites is an important piece of information that is needed to carry out the LCC of the repair business model of a manufacturer using CFRP-ff or CFRP-bio in its production system. From the laboratory data, it is evident that the costs of the two new CF composites are higher than the conventional ones. Thus, proper cost estimation of the new material is critical. In this study, the top-down approach is applied to calculate the relative costs of CFRP-ff and CFRP-bio. The rationale behind using this method is the fact that there are uncertainties for cost analysis of the new CF composites that do not allow a detailed level cost analysis of the product. These uncertainties arise from the fact that these materials are developed in the laboratory, where the material costs are much higher and not comparable to the industry costs. For the top-down method, the expert judgement is used to calculate the relative costs of CFRP-ff and CFRP-bio. Three experts were selected and interviewed to evaluate the relative cost of the new CFRPs. A production manager, technical manager and the senior R&D manager of the case study company are all experienced in composite materials’ formulation and production and composite production technologies and processes. To estimate the industrial cost of the CFRP-ff and CFRP-bio (prepreg), the first step was to add up the costs of the production material and manufacturing as the initial building blocks of the total cost. Given that the production procedure of the CFRP-ff and CFRP-bio is the same as the conventional procedure, the manufacturing cost of the new materials was considered similar to that of the conventional materials. In other words, the main difference between newly developed CFRP and the conventional one lies in the material cost. Using the top-down method first, the relative material cost of the CFRP-ff and CFRP-bio is compared to the price of conventional CFRP. According to composite experts, the relative material cost of the CFRP-ff at the industrial scale can be 30% higher than the price of a conventional one with the coefficient of −20% to +20%. In other words, a 30% cost increase in the CFRP cost is considered as the base case, while the best case is where material increases by 10%, and the worst case is when a material cost increases by 50%. On the other hand, as the material cost of CFRP-bio is more expensive than the CFRP-ff, experts’ judgment indicated that the relative material cost for CFRP-bio can be 50% higher than the conventional CFRP with the coefficient of −20% and +20%. To estimate the final production cost of the new CFRP in addition to the material cost, it also important to know the cost breakdown structure of the CFRP. According to Graf et al. [89], in CFRP components, the material cost accounts for 36%, followed by 64% for the manufacturing cost. Based on this cost structure, when the material cost of CFRP-ff increases by 10%, 30% or 50%, the total cost of CFRP increases by 3.6%, 10.8% and 18%. Concerning CFRP-bio, when the material cost increases by 30%, 50% and 70%, the total cost increases by 10.8%, 18% and 25.2%.In this study, a net present value (NPV) and discounted payback period of the investment are used to evaluate the economic performance of a system. The discount rate is assumed to be 10%, and the NPV is calculated over a period of 10 years.

#### 2.2.5. Sensitivity Analysis

Sensitivity analysis is used to investigate how the output is affected by the system’s uncertainty. As a result, a sensitivity analysis is performed with assumptions on the parameters with higher uncertainty [62,90]. In this study, the following steps were adopted to carry out the sensitivity analysis:Identification of the uncertain parameters;Definition of the potential ranges for the parameters resulting in various scenarios;The economic assessment of producer and thermal recycler carried out through feeding different value parameters into the economic model.

Based on the steps mentioned above, in the following, the most uncertain parameters that were used to assess the profitability of producer and recycler business models, and their ranges are presented. Concerning the repair business model of the producer, among various input data affecting the LCC, the relative cost of new materials was identified as the most uncertain parameter as it has the highest impact on the NPV and payback period. Table 13 and Table 14 present the variability of the parameter, resulting in three scenarios for each type of composite. The investment cost is not considered as a variable parameter since it is not affected by uncertainty, and we could rely on the result of the economic system. In addition, the repair cost is not also considered as an uncertain parameter since this cost is relatively low and has no impact on the financial output of the economic indicator. In this regard, if the repair cost increased by 1000%, the NPV would increase by 0.24%, representing that the impact of this parameter on the final results is almost negligible.

Regarding the thermal recycler business model, among various input data affecting the LCC, the following parameters were used to conduct the sensitivity analysis: the annual amount of CFRP waste, the selling price of recycled CF (rCF) and the transportation cost to deliver the waste to the recycler’s plant. These parameters were affected by a considerable level of uncertainty. To capture the uncertainties surrounding the recycled CF selling price and the transportation cost, experts provided us with ranges for these parameters as it is presented in Section 2.2.3. The amount of waste that the recycler can receive each year is considered a variable since this parameter can have a high impact on the NPV and payback period. The amount of waste that the recycler can receive each year was hypothesized as a percentage of the saturation rate of the plant: 75%, 85%, 90%, 95% and 100%. Given that the capacity of the plant is 250 ton/year, the potential ranges of the waste are presented in Table 15. The combination of these three parameters with their ranges led to 45 scenarios.

## 3. Results and Discussion

### 3.1. LCA Results

This section provides preliminary results of a new technology still under development at the laboratory scale in order to identify potential opportunities to be further developed at the industrial level [31]. This section shows the variability across scenarios of the following three challenges of composites in the circular economy perspective: use of bio-based materials, repairing and recycling. Table 16 shows the LCIA of the FU of the study for the six scenarios performed across the eighteen environmental categories of the ReCiPe methodology listed in Section 2.1.3. Scenario A1 shows the highest results in almost all the impact categories, except for *Fine particulate matter formation—PM, Freshwater ecotoxicity—FEX, Freshwater eutrophication—FE, Land use—LU* and *Stratospheric ozone depletion—SOD*. The highest value for *Land use—LU* is shown in scenario B1, meaning that the variability of this category is more related to differences in the curing agent of the two composites. The lowest value is shown for scenario A3. Scenario B2 and B3 show the lowest values for several impact categories, meaning that a combination of bio-based materials, repairing and recycling could result in a reduction in the environmental impacts. The highest values for *Fine particulate matter formation—PM, Freshwater ecotoxicity—FEX, Freshwater eutrophication—FE* and *Stratospheric ozone depletion—SOD* are shown in scenario A3, with slightly lower values for scenario B3. Those results are mainly related to the thermal process of recycling and in particular to the high energy intensity of the process. Normalization of the LCA results is provided in Appendix A.

The contribution of the environmental impacts across the life stages of the product is shown in Figure 3. *CF production* and *Manufacturing* contribute to the entire life cycle for scenarios A1 and B1 in the range of 29% and 49% for all the impact categories, except for *Marine Eutrophication* of scenario A1, which is equal to 9%. This result is mainly due to the several contributions of *substitution* of the composite. The contribution of those two processes further increases considering scenarios with repairing (i.e., A2, A3, B2 and B3) being in the range of 42% and 98% for almost all the impact categories, except for *Marine Eutrophication* of scenarios A2 and A3, which is equal to 19%, and for *Fine particulate matter formation—PM, Freshwater ecotoxicity—FEX, Freshwater eutrophication—FE* and *Stratospheric ozone depletion—SOD* of scenario A3 and B3, mainly due to the energy intensive process of thermal recycling. For those categories, the thermal process covers 86%–99% of the impacts. The other EoL treatments have low contributions compared to the overall life cycle stages. However, further clarification on different EoLs is shown below in this section. *Fossil-fuel-based curing agent synthesis* and *bio-based curing agent synthesis* show similar contributions across scenarios for almost all impact categories of between 1.8% and 13%. The exception is shown for *Marine Eutrophication* of groups in scenario A due to the higher contribution to the overall life cycle stages being in the range of 40% and 80%.

Focusing on *Global warming* category, Figure 4 shows the contribution of each life cycle stage across the six scenarios. Excluding the EoL phase of composites, *Manufacturing* and *CF production* phases cover the majority of the CO${}_{2}$eq emissions (from 42% up to 86% of the total). However, their contribution is the same across the scenarios since the process is the same. The synthesis of both curing agents have different contributions (from 4% to 13%), mainly due to the substitution. The substitution of the composite results to double the environmental impact in the overall life cycle of the product. Indeed, the substitution covers the majority of the impact since a second product with the same properties and manufacturing pathway is required. Focusing on the curing agents, the substitution of the fossil-fuel-based curing agent with the bio-based curing agent reduces the impact by up to 40%. Finally, the recycling of the product avoids the production of CFs and epoxy resin from virgin material, reducing the potential environmental impacts by 18–37%. The negative bar represents the avoided primary production of CFs and epoxy resins due to the recycling of CFRP waste. Differences in the negative impacts of recycling is due to different efficiencies of recycling both materials.

As concerns the use of alternative bio-based materials, the blue bars in Figure 5, which refer to scenarios A1 and B1, show that CFRP-ff and CFRP-bio have similar impacts across almost all the environmental categories. The substitution of the dynamic fossil-fuel-based curing agent with the bio-based material slightly reduces the impact for almost all the categories. A more relevant reduction is shown for the *Marine eutrophication* impact category. Indeed, the synthesis of the fossil-fuel-based cured epoxy resin covers 40% of the overall impact of this category. Furthermore, 99% of the impact of the curing agent synthesis is represented by the *meta-phenylene diamine* production—the proxy material used to model the dynamic curing agent. Sensitivity analysis on curing agents modeling are described in Section 3.1.1. CFRP-bio shows a slightly higher impact in the *Land use* impact category, mainly due to the pre-treatment of the epoxy resin mixing with the catalyst and the curing agent.

As concerns the repairing of damages in the composite, Figure 5 shows a clear reduction in the impacts in almost all the environmental categories in the case of repairing the composite (scenario A2 and B2) instead of substituting the composite (scenario A1 and B1). Repairing the composite results in a reduction in the impact of between 30% and 50% for fifteen impact categories. *Freshwater, Marine and Terrestrial ecotoxicity* are the environmental categories with a minor reduction in the impacts due to reparation compared to substitution (in the range of 4% and 18%) because of the laser machine contribution. Figure 5 shows little difference in results between the scenario A group and the scenario B group. This outlines that differences in the curing agents slightly affect the final results. It is noted that the comparison between scenarios with substitution and scenarios with repair excluded the EoL evaluation in order to compare only the impact variation due to the repair.

As concerns the recycling, Figure 6 shows the results regarding the three scenarios considered for both composites: landfill and incineration, mechanical recycling and thermal recycling. As a reference, the scenarios with the highest value were set as 100%, and the other scenarios were scaled accordingly. Comparing landfill/incineration with mechanical recycling, across the eighteen environmental categories, the impact decreases from 21% in the *Mineral resource scarcity* category up to 99% in the *Stratospheric ozone depletion* category. Comparing landfill/incineration with thermal recycling, the results are less stable. Indeed, thermal recycling shows the highest values for nine impact categories because of the high intensity of the thermal process. However, the thermal recycling can recover CFs and epoxy resin. This comparative assessment between different EoL treatments was not performed considering the avoided production of new materials due to, respectively, mechanical and thermal recycling. Indeed, from a circular economy perspective, both recycling treatments reduce the request of new raw materials, which does not occur in the case of landfills and incineration. Further assessments should include the additional production of CF and epoxy resin that is instead recovered during mechanical and thermal recycling. The negative impact values of *rCF and rEpoxy* shown in Figure 4 are due to the avoided impact of virgin CF and epoxy resin because of their recycling, which are related to the chosen recycling option and the efficiency of recovery. As a matter of fact, the pyrolysis treatment recovers 99% of CF and 85% of epoxy resin, while mechanical recycling recovers 60% of the CFs.

Summarizing the obtained results, LCIA shows that CF production and manufacturing of the product through autoclave technology are the most relevant processes in the entire life cycle of the product. Indeed, as shown in Figure 3, in scenarios with substitution (i.e., A1 and B1), both processes contributes to the entire life cycle in the range of 29% and 49% for all the impact categories, except for *Marine Eutrophication* of scenario A1, which is equal to 9%. The contribution of those two processes further increases considering scenarios with reparation (i.e., A2, A3, B2 and B3) to the range of 42% and 98%, except for *Marine Eutrophication* of scenarios A2 and A3, which is equal to 19%, and for *Fine particulate matter formation—PM, Freshwater ecotoxicity—FEX, Freshwater eutrophication—FE* and *Stratospheric ozone depletion—SOD* of scenario A3 and B3, mainly due to the energy intensive process of thermal recycling. Although comparison with other LCA studies on composites is not always possible due to differences in the FU, these results are aligned with other studies [49,58,61,72]. According to these results, efforts to identify alternative fibers to substitute CF have also been made by other studies [30,61,74]. However, the present case study did not include alternative bio-based fibers since laboratory tests on the possibility of repairing and recycling were not performed. Further research could be conducted in studying the possibility of repairing and recycling these bio-based fibers. Limitations were also due to *Production of the prepreg*, which was based on laboratory data that were scaled up to industrial scale. Those data should be further updated with primary data on pilot tests in order to verify the robustness of this process. It is noted that the analysis did not include the use phase of the product because it was assumed that the impact to be allocated to a small component compared to the entire car was limited. However, contributions of the use phase could change according to the product considered in the study and its relevance in the use phase. Indeed, Roy et al. [61] outlined that the use phase of a car component covers the majority of the impacts of the overall life cycle.

Across the scenarios developed, B3 included the substitution of the fossil-fuel-based curing agent with the bio-based curing agent, the reparation of the composite and the pyrolysis through thermal recycling of the composite at the EoL. As shown in Figure 5, B3 is the scenario with the highest environmental benefit among the impact categories compared to the baseline scenario (i.e., A1). This result outlines that the three above-mentioned drivers of the circular economy perspective could provide a reduction in the environmental impact from 10% up to 86% depending on the category considered. However, the highest benefits in terms of environmental burdens could be obtained with the implementation of repairing procedures on an industrial scale with a reduction up to 44–50% for eleven impact categories. Indeed, the analysis of environmental impacts due to the substitution of a damaged product to guarantee the entire life cycle of a service, for instance the car life cycle, is not usually included in LCA studies. Furthermore, substitution could occur also for aesthetic damages that do not compromise the structural properties of a product.

Finally, the lowest environmental benefits are observed substituting the traditional material with the bio-based material, which results in a variation in the range of −9% and +10%. According to the results obtained in this LCA, repairing is the driver with the highest environmental benefits compared to recycling and using alternative bio-based materials.

#### 3.1.1. Sensitivity Analysis of LCA

Uncertainties on LCA results are mainly due to (1) primary data from the laboratory scale that were scaled-up to the industrial scale with assumptions and (2) the absence of literature data of some materials modeled and further modeling according to proxy data provided by the chosen database. Thus, sensitivity analysis was performed within this context to evaluate the robustness of the chosen modeling and how the results are affected by modeling assumptions. Furthermore, a sensitivity analysis was performed on the baseline scenario (i.e., A1), assuming that the damaged product is sent to mechanical or thermal recycling. As a matter of fact, the following sensitivity analysis were performed:Fossil-fuel-based dynamic curing agent modeling;Electrodes modeling for mechanical recycling;Different EoLs of the damaged product in scenario A1.

The fossil-fuel-based dynamic curing agent was modeled with *market for meta-phenylene diamine|meta-phenylene diamine|Cutoff*. As explained in Section 2.1.2, meta-phenylene diamine was chosen as the aromatic amine in accordance with the literature [75,76]. As shown in Figure 5, the two chosen types of curing agents (i.e., fossil fuel and bio-based) showed a similar impact for all the impact categories, except for *Marine eutrophication*. The following two proxy materials were modeled in order to verify the trend: *market for diethanolamine|diethanolamine|Cutoff* and *market for diethanolamine|diethanolamine|Cutoff*. Variations in the results were observed to be in the range of -10% and +3% with respect of the baseline scenario (i.e., A1). Furthermore, in both cases, *Marine eutrophication* decreased by up to 20% compared to scenario A1, resulting in a similar value as CFRP-bio. Sensitivity analysis on the fossil-fuel-based curing agent modeling showed that the chosen model did not significantly vary with respect to other available options of modeling. However, this evaluation is limited to available datasets, and further research on curing agents modeling could change the results.

The use of electrodes was modeled with *market for electrode, negative, LiC6|electrode, negative, LiC6|Cutoff*. Due to the absence of information on the electrode properties, uncertainties of the modeling were analyzed. Two more scenarios of mechanical recycling were developed, substituting the modeling of electrode production with *market for electrode, negative, Ni|electrode, negative, Ni|Cutoff* and *market for electrode, positive, LaNi5|electrode, positive, LaNi5|Cutoff* in order to verify the robustness of the modeling. Figure 7 shows high variability in the results for *Fine particulate matter formation*, *Stratospheric ozone depletion*, *Terrestrial acidification* and *Terrestrial ecotoxicity*. However, *Freshwater eutrophication*, *Global Warming*, *Ionizing radiation* and *Marine eutrophication* are the impact categories with the lowest variability across the three possible models. Sensitivity analysis on electrodes used for mechanical recycling showed significant variation among the possible alternative datasets for some impact categories (Figure 7). Further research on the used electrodes could enhance the robustness of the data.

An alternative case was analyzed in order to evaluate the weight of the recycling in the case of substitution. This sensitivity analysis aimed to evaluate how the results might vary according to the selected EoL in case substitution is still necessary. Instead of sending the damaged product to a landfill or for incineration, this sensitivity analysis assumes that the damaged product is sent to mechanical recycling. The analysis comprises the production of CF, epoxy resin and prepreg, the manufacturing of the paddle shifters and the substitution. However, the contribution is almost the same as the base case (i.e., substitution without recycling). This means that the EoL treatment has a minimum weight compared to the rest of the life cycle stages. However, a reduction in the impact can be observed due to the mechanical recycling of the composite if further analysis of the avoided production of new CFs is considered.

### 3.2. LCC Results

In this section, the LCC results of the repair producer and thermal recycling businesses of a recycler are reported.

#### 3.2.1. Repair Business Model of a Producer

The results of the economic assessment of the repair business model of the producer when the product is made from CFRP-ff are demonstrated in Table 17. As can be observed, the financial modeling indicates that for all three scenarios, with different ranges of CFRP-ff costs, the manufacturing and repair business model of a producer is always profitable as NPV is positive. Moreover, there is limited variability (0.8%) across the scenarios concerning NPV. The discounted payback period for all scenarios is achieved within the fifth year. Moving next to Table 18, the result of the economic assessment of a producer using CFRP-bio is demonstrated. The financial results show that this business is also profitable with limited variability (0.8%) across the scenarios with respect to NPV, and the discounted payback period is also achieved in the fifth year. Here, it is also essential to emphasize that the investment cost for the laser is high, whereas the sale volume of this product is relatively low, particularly in the context of automotive, where sale figures are high. Therefore, although the payback period is achieved within the fifth year, the payback period decreases if the sale volume increases.

To clearly understand why NPVs of all scenarios are positive, it is important to highlight that by manufacturing products from CFRP-ff, the total cost of the product increases by 0.08% to 0.39% compared to the total cost of the product built from the conventional CFRP. Concerning the product made from CFRP-bio, the total cost of the product increases by 0.23% to 0.55% compared to the total cost of the conventional CFRP product. Moreover, the repair cost is relatively low, 0.05%, compared to the total production cost. Therefore, it is clear that product repair can provide save a lot of costs as the producer can recapture the monetary value of the manufacturing process. The economic assessment of a product built from the new CF composites (CFRP-ff and CFRP-bio) and taking advantage of the reparability of the material to address the defective products shows that, overall, the manufacturing coupled with the repair business is very promising. In the current context, when the CF composite product is damaged, the manufacturer loss includes the entire production cost (both material and manufacturing cost will be discarded). By integrating the repair facilities into the manufacturing plant and bearing a relatively low repair cost, the manufacturer would avoid their loss of the whole production process. Therefore, the reparability of the new material is a significant opportunity for manufacturers to save money.

#### 3.2.2. Thermal Recycling for a Recycler

The economic assessment results of a thermal recycler are summarized in Figure 8. The results display a positive NPV (10 years) for 4% of scenarios (2/45 scenarios) and show significant variability across the 45 scenarios. The positive NPV is achieved when the recycled CF price is at the maximum value (6 EUR/kg), and the transportation cost is at the minimum value (0.15 EUR/kg), and the plant is at least 95% saturated. The average discounted payback period for the two scenarios showcasing a positive NPV is obtained within 9.5 years. The sensitivity analysis shows that recycled CF prices are the most significant parameter impacting the NPV and discounted payback period. Moreover, the transportation cost is also the second most impactful parameter on both the NPV and the discounted payback period. Finally, a secondary element affecting the profitability of results is the annual amount of CFRP waste.

The economic sustainability of the thermal recycler that receives CFRP-ff/bio waste indicates that the business can be profitable only under a specific condition. Therefore, the economic results are affected by uncertainties that arise from the market price of recycled CF, the transportation cost, and the amount of waste. In this regard, a recycler cannot control the price of the recycled CF since the variabilities associated with this parameter are caused by the adverse economic cycle, uncertain financial stability and low demand for products at the global level [91,92]. However, a recycler can reduce the variabilities associated with the transportation cost and amount of CFRP waste. From a business point of view, strategic management of the reverse supply chain is critical for the profitability of the business of a thermal recycler. Strategic partnerships with organizations operating in the reverse supply chain, such as compliance schemes or waste collectors, are potential pathways to keep transportation costs minimal. In addition, such partnership can also guarantee a steady supply of a certain volume of waste, which is crucial for the successful implementation of the business.

## 4. Conclusions

The present study attempted to provide an LCA coupled with an LCC of a new CFRP developed in a laboratory that could be repaired and recycled. The scope of this LCA was to analyze in a single study the following three relevant drivers that could enhance a circular economy: repairing, recycling and using bio-based materials. Firstly, according to the LCA results, the substitution of a CFRP product can double the overall environmental impact in the entire life cycle of a service. The introduction of repairing the product could reduce the environmental impacts from 17% to 50% according to each category (Figure 5). Furthermore, the substitution of the composite product may also occur for aesthetic damages, for instance, in a luxury car, which means that the damaged product becomes waste, even though its mechanical properties are not compromised. Nevertheless, the substitution of a CFRP product is not usually included in LCA studies, and hence, the environmental and cost effects of manufacturing an additional product are not quantified. Indeed, damages to a product and its subsequent substitution is unexpected and not desired considering the entire life cycle of a product. Secondly, the recovery of rCf and epoxy resin through mechanical and thermal recycling allowed reducing the environmental impact related to the waste management up to 99% for some categories (Figure 6). Finally, only the curing agent was analyzed as a bio-based material in the composite (group of scenarios B) in order to guarantee the properties of the developed composite to be repairable. Hence, the curing agent is a small percentage if compared to the overall weight of the composite. As a result, the two materials (i.e., CFRP-ff and CFRP-bio) had similar environmental impacts across the eighteen impact categories (Figure 5). The results may change increasing the percentage of bio-based materials to produce a CFRP composite.

From an economic point of view, the LCC assessment showed that repair technologies and processes are highly convenient for composite parts’ manufacturers due to the typically high defect rate of composites and the high reproduction costs that have to be sustained. Under these conditions, investing in laser-based technology and the higher cost of the new repairable composite materials that should be sustained are largely covered by cost savings in short time periods. Regarding the thermal recycling of the newly assessed material and technologies, it appeared that the business can be economically sustainable only if the recovered CF is sold at its maximum price, the transportation cost is at its lowest and the recycler receives high volumes of waste. In this regard, this business is characterized by significant uncertainties, which can be reduced through a vertical integration or strategic partnerships with actors involved in the reverse supply chain in order to guarantee big treatment volumes and a stable transportation cost.

The LCA and LCC results obtained are affected by uncertainties due to the absence of data or scaling up of laboratory data to the industrial scale, as analyzed in Section 3.1.1 and Section 2.2.5. The LCA and LCC allowed collecting preliminary environmental and costing information that is crucial for further developing the product at pilot and industrial scales. Further research on those two types of composites and repairing technologies with a pilot test are fundamental to obtaining a new type of product that can be repaired, which could push on the LCA and LCC’s robustness. Moreover, performing LCA and LCC during all those steps could be crucial to both understanding how the results change according to data availability and production scale (i.e., from laboratory to industrial scale) and monitoring the environmental and costing efficiency as decision support tools for its development.

## Figures and Tables

**Figure 1 materials-15-02986-f001:**
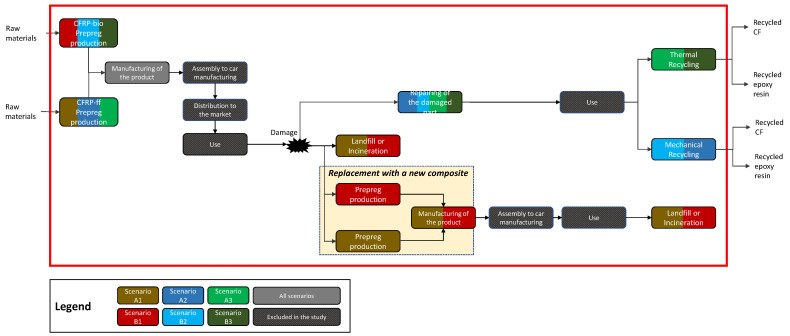
System boundaries of each scenario of the comparative LCA.

**Figure 2 materials-15-02986-f002:**
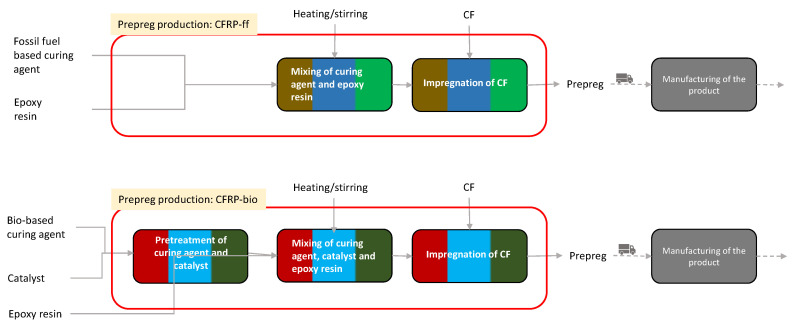
Prepreg production process on a laboratory scale. CFRP-ff (i.e., group of scenario A) includes two main steps: firstly, mixing the dynamic curing agent with the epoxy resin, and secondly, the impregnation of the CF. CFRP-bio (i.e., group of scenario B) includes the same steps of scenario A plus two additional pretreatment steps needed for the dynamic curing agent and catalyst. Colors represent the different scenarios.

**Figure 3 materials-15-02986-f003:**
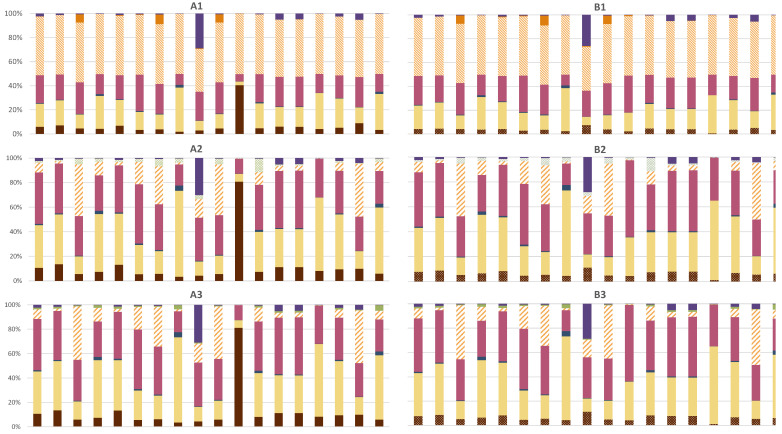
Impactcontribution of the life cycle stages of the product across the impact categories analyzed for the six scenarios. *CF production* and *Manufacturing* cover the majority of the impact. This contribution decreases for scenario A1 and B1 because of the *substitution*. Similar contributions are shown for the synthesis of both curing agents for almost all categories. Limited contributions are observed for *Prepreg*. Thermal recycling is the only EoL option that covers the majority of the impact for three categories (i.e., A3 and B3).

**Figure 4 materials-15-02986-f004:**
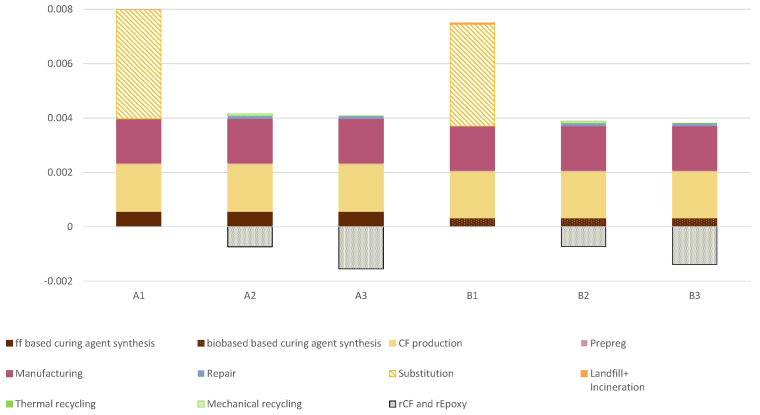
Impact contribution for the 6 scenarios across each life cycle of the FU for the *Global Warming* impact category. Negative impact values of *rCF and rEpoxy* are due to the avoided impact of virgin CF and epoxy resin because of their recycling. The quantitative avoided impacts are related to the chosen recycling option and its efficiency of recovery.

**Figure 5 materials-15-02986-f005:**
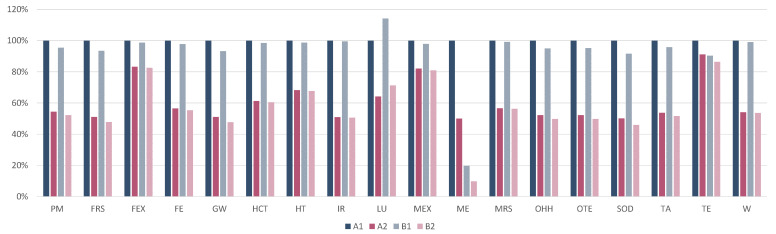
Comparative LCA between CFRP-ff with substitution (scenario A1), CFRP-ff with repair and mechanical recycling (scenario A2), CFRP-ff with repair and thermal recycling (scenario A3), CFRP-bio with substitution (scenario B1), CFRP-bio with repair and mechanical recycling (scenario B2) and CFRP-bio with repair and thermal recycling (scenario B3). Repairing the composite reduces the impact by up to 50% compared to scenarios with substitution. Less reduction in the impact when repairing are shown for the *Freshwater, Marine and Terrestrial Ecotoxicity* impact categories. The substitution of the curing agent slightly contributed to reducing the impacts among all the categories. The same occurs comparing mechanical and thermal recycling.

**Figure 6 materials-15-02986-f006:**
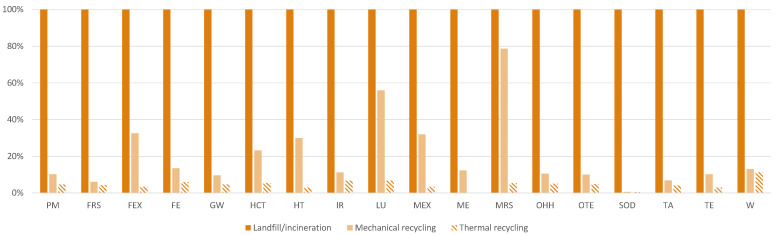
Comparative LCA between EoLs of composites. Incineration and landfill are representative of the most common composite waste management and hence are the baseline of the analysis. Relevant reduction in all the impact categories both for the thermal and mechanical recycling pathways. Limited environmental benefits compared to the baseline are shown for mechanical recycling in *Mineral resources scarcity*.

**Figure 7 materials-15-02986-f007:**
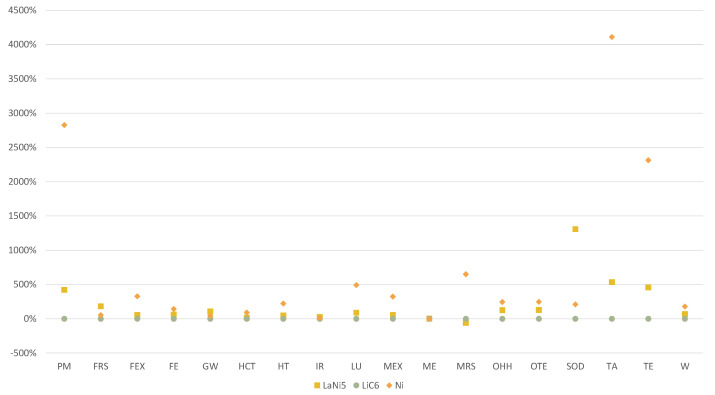
Sensitivity analysis on mechanical recycling. According to different providers of electrode production, some of the impact categories are affected by the results. Compared to LiC6, which is the base case, the highest and the lowest variability is observed, respectively, for *Terrestrial acidification* and *Marine eutrophication* categories.

**Figure 8 materials-15-02986-f008:**
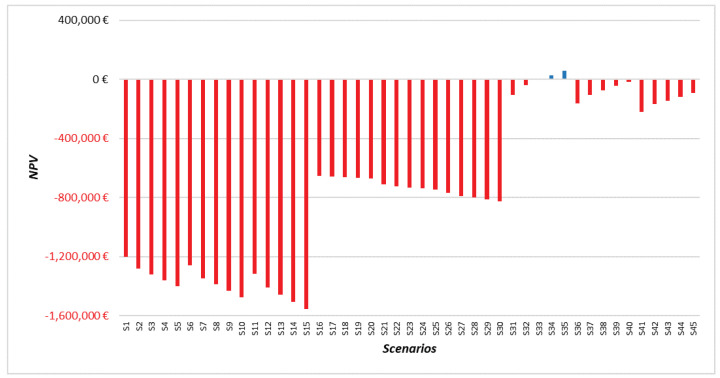
NPV of the thermal recycler based on various scenarios. The three parameters that affect the NPVs are the selling price of the recycled CF, the transportation and the volume of waste. The NPV is positive when the CF selling price is at a maximum, the transportation is at a minimum and the plant is saturated at least by 95%.

**Table 1 materials-15-02986-t001:** Description of the scenarios of the comparative LCA.

Number of Scenario	Name of Composite	Composition	Repairing	End-of-Life
A1	CFRP-ff	curing agent 1+epoxy resin+CF	No	50% Landfill + 50% Incineration
A2			Yes	Mechanical recycling
A3			Yes	Thermal recycling
B1	CFRP-bio	curing agent 2+epoxy resin+CF	No	50% Landfill + 50% Incineration
B2			Yes	Mechanical recycling
B3			Yes	Thermal recycling

**Table 2 materials-15-02986-t002:** Differences in primary data available between laboratory and industrial scales. For each production process, specific data are needed as input to build the LCA model. On the laboratory scale, data on production processes are, for instance, temperature, duration of the process, pH, etc. However, those inputs are not directly usable in an LCA study. On the industrial level, the available data are directly usable in an LCA, such as energy and auxiliary material consumption.

Production Process	Laboratory Scale (Unit = [g])	Industrial Scale (Unit = [kg or t])
Heating	Temperature, Duration	Energy and auxiliary material consumption
Stirring	Temperature, Rotation velocity, Duration	Energy and auxiliary material consumption
Grinding	Duration, Dimension of particles	Energy and auxiliary material consumption
Filtration	Auxiliary materials, Duration	energy and auxiliary material consumption

**Table 3 materials-15-02986-t003:** Calculation methodology adopted from Piccinno et al. [46] to perform the scale up of the energy consumption from laboratory scale to industrial scale. *C${}_{p}$* = Specific heat capacity [J/(kg*K)]; *m${}_{mix}$* = mass of the reaction mixture (kg); *T${}_{r}$* = temperature of reaction (K); *t* = time of reaction (s); $\rho_{mix}$ = density of the mixture (kg/m).

Production Process	Calculation
Heating	$Q_{react}=\left[C_{p}*m_{mix}*\left(T_{r}-298.15K\right)+3.303*\left(T_{r}-298.15K\right)*t\right]/0.75$
Stirring	$E_{stir}=0.018\rho_{mix}*t$
Grinding	8–16 kWh/ton
Filtration	1–10 kWh/ton

**Table 4 materials-15-02986-t004:** Mixing of the epoxy resin and the dynamic curing agent according to the two scenarios. Data are referred to 1 kg of uncured vitrimer. Epoxy resin/dynamic curing agent ratio from de Luzuriaga et al. [3].

Input	CFRP-ff	CFRP-bio
Epoxy resin (kg)	0.7	0.70
Fossil fuel based curing agent (kg)	0.3	-
Bio-based curing agent (kg)	-	0.27
Catalyst (kg)	-	0.03
Electricity (kWh)	5.9 × 10${}^{-2}$	1.3 × 10${}^{-1}$
Steam (kg)	-	33.4
Cooling water (kg)	-	0.59
Other auxiliary materials (kg)	-	21.8

**Table 5 materials-15-02986-t005:** Impregration of the CF with the epoxy resin. Data refer to 1 kg of prepreg. Data on energy consumption were taken from Song et al. [54].

Input	CFRP-ff	CFRP-bio
Epoxy resin (kg)	0.51	0.52
Carbon Fiber (CF) (kg)	0.49	0.48
Electricity (MJ)	40	40

**Table 6 materials-15-02986-t006:** Inventory data for the manufacturing of the paddle shifters taken from Forcellese et al. [72].

Material	Quantity (kg)
*Input*	
Prepreg (CFRP-ff or CFRP-bio)	15.7
Electric energy (kWh)	17.074
Master mold in polyurethane foam	30.5
Mold (in CFRP)	0.095
Polyamide 66 (PA66)	0.5
Polyethylene terephthalate (PET)	0.375
Polytetrafluoroethylene (PTFE)	0.055
Organic solvent	0.03
*output*	
Paddle shifters	11
scraps of prepreg	4.7
scraps of master mold	9.5

**Table 7 materials-15-02986-t007:** Inventory data for the mechanical recycling of paddle shifters. The consumption of the electrodes were allocated to 1 kg of CFRP waste on the mass basis. The data are valid both for CFRP-ff and CFRP-bio.

Material	Quantity
*Input*	
Paddle shifters waste (kg)	1
Electric consumption (kWh)	1
electrodes (item)	0.005
Water (L)	5
*output*	
recycled CFRP (kg)	0.6
waste CFRP (kg)	0.4
Water losses (L)	0.25

**Table 8 materials-15-02986-t008:** Inventory data for CO${}_{2}$-assisted pyrolysis of the paddle shifters. The CO${}_{2}$ was not included in the mass balance since it is recirculated in the process. CO${}_{2}$ losses were not considered. The data are valid both for CFRP-ff and CFRP-bio. Data of electricity consumption were taken from Nunes et al. [52].

Material	Quantity
*Input*	
Paddle shifters waste (kg)	1
Electric consumption (MJ)	54
*output*	
CFs recycling efficiency (%)	0.99
Epoxy resin recycling efficiency (%)	0.85
solid residue (kg)	0.25

**Table 9 materials-15-02986-t009:** Inventory data of transportation.

Transportation	Distance (km)
CF to prepreg production site	20,145
Fossil-fuel-based epoxy resin to prepreg production site	10,130
Bio-based epoxy resin to prepreg production site	10,130
Prepreg production to paddle shifter manufacturing site	1600
CFRP waste to landfill/incineration	50
CFRP waste to repairing site	150
CFRP waste to mechanical recycling	1000
CFRP waste to thermal recycling	1000

**Table 10 materials-15-02986-t010:** Input data of manufacturing cost of the paddle shifter.

Data	Unit of Measure	Value
Yearly product volume	piece/year	800
Cost of conventional CFRP (prepreg)	EUR/kg	50
Profit margin	%	50
Defect rate	%	20
Material cost compared to the total product cost	%	2.2
Manufacturing cost compared to the total product cost	%	97.8

**Table 11 materials-15-02986-t011:** Characteristics and cost of the repair technology. Electricity power cost were taken from [87]. Personnel wages data were taken from [88].

Data	Unit of Measure	Value
Investment cost	EUR	200k
Yearly maintenance cost of laser facility	EUR	10k
Dimension of damage area	cm${}^{2}$	1
Electricity consumption to repair the damage area	kWh	0.111
Electrical power cost	EUR/kWh	0.1254
Personnel wages	EUR/h	28
Time needed to repair the damage area	seconds	5

**Table 12 materials-15-02986-t012:** Characteristics and costs related to thermal recycling. Electricity power costs were taken from [87]. Personnel wages data were taken from [88].

Data	Unit of Measure	Value
Investment cost of thermal recycling facilities	EUR	550k
Capacity of plants	ton/year	250
Working days per year	day	240
Working hours per day	hour	8
Electrical power cost	EUR/kWh	0.1254
Personnel wages	EUR/h	28
Overhead and administration cost	%	20
Maintenance time (after every cycle of 8 h)	hour	1
Gas cost (CO${}_{2}$)	EUR/ton	150
Transport cost	EUR/kg	0.2
Recycled CF price	EUR	5

**Table 13 materials-15-02986-t013:** Parameters used for sensitivity analysis of the repair business of paddle shifters made from CFRP-ff. The conventional CFRP cost is 50 EUR/kg.

Range	Relative Cost Increase Compared to the Conventional CFRP(%)	Cost of CFRP-ff (EUR/kg)
R1	3.6%	51.8
R2	10.8%	55.4
R3	18%	59

**Table 14 materials-15-02986-t014:** Parameters used for sensitivity analysis of the repair business of paddle shifter made from CFRP-bio. The conventional CFRP cost is 50 EUR/kg.

Range	Relative Cost Increase Compared to the Conventional CFRP(%)	Cost of CFRP-bio (EUR/kg)
R1	10.8%	55.4
R2	18%	59
R3	25.2%	62.6

**Table 15 materials-15-02986-t015:** Parameters used for sensitivity analysis of the thermal recycling business of a recycler.

Ranges	Price of Recycled CF (EUR/kg)	Transportation Cost (EUR/kg)	Annual Amount of Waste (ton/Year)
R1	4	0.15	187.5
R2	5	0.20	212.5
R3	6	0.25	225
R4	-	-	237.5
R5	-	-	250

**Table 16 materials-15-02986-t016:** LCA results of the six scenarios performed across the impact categories. symbol (+) shows the highest value compared to the other scenarios; symbol (−) shows the lowest value compared to other scenarios. The visualization shows A1 has the highest results for almost all categories, and B2 and B3 have the lowest values in several impact categories.

Impact Category	Unit	A1	A2	A3	B1	B2	B3
PM	kgPM${}_{2.5}$eq	1.20 × 10${}^{-5}$	6.66 × 10${}^{-6}$	5.43 × 10${}^{-5}$ (+)	1.15 × 10${}^{-5}$	6.39 × 10${}^{-6}$ (−)	5.41 × 10${}^{-5}$
FRS	kgoileq	2.67 × 10${}^{-3}$ (+)	1.38 × 10${}^{-3}$	1.89 × 10${}^{-3}$	2.50 × 10${}^{-3}$	1.29 × 10${}^{-3}$ (−)	1.80 × 10${}^{-3}$
FEX	kg1,4-DCB	3.89 × 10${}^{-4}$	3.16 × 10${}^{-4}$	4.60 × 10${}^{-4}$ (+)	3.84 × 10${}^{-4}$	3.14 × 10${}^{-4}$ (−)	4.57 × 10${}^{-4}$
FE	kgPeq	2.89 × 10${}^{-6}$	1.68 × 10${}^{-6}$	1.17 × 10${}^{-5}$ (+)	2.83 × 10${}^{-6}$	1.65 × 10${}^{-6}$ (−)	1.17 × 10${}^{-5}$
GW	kgCO${}_{2}$eq	8.13 × 10${}^{-3}$ (+)	4.21 × 10${}^{-3}$	4.14 × 10${}^{-3}$	7.60 × 10${}^{-3}$	3.94 × 10${}^{-3}$	3.87 × 10${}^{-3}$ (−)
HCT	kg1,4-DCB	3.51 × 10${}^{-4}$ (+)	2.19 × 10${}^{-4}$	3.34 × 10${}^{-4}$	3.46 × 10${}^{-4}$	2.17 × 10${}^{-4}$ (−)	3.32 × 10${}^{-4}$
HT	kg1,4-DCB	6.54 × 10${}^{-3}$ (+)	4.37 × 10${}^{-3}$	4.19 × 10${}^{-3}$	6.47 × 10${}^{-3}$	4.33 × 10${}^{-3}$	4.15 × 10${}^{-3}$ (−)
IR	kBqCo-60eq	1.02 × 10${}^{-3}$ (+)	5.37 × 10${}^{-4}$	5.29 × 10${}^{-4}$	1.02 × 10${}^{-3}$	5.35 × 10${}^{-4}$	5.26 × 10${}^{-4}$ (−)
LU	m${}^{2}$a crop eq	6.99 × 10${}^{-6}$	4.76 × 10${}^{-6}$	4.71 × 10${}^{-6}$ (−)	7.70 × 10${}^{-6}$ (+)	5.12 × 10${}^{-6}$	5.06 × 10${}^{-6}$
MEX	kg1,4-DCB	5.09 × 10${}^{-4}$ (+)	4.08 × 10${}^{-4}$	3.89 × 10${}^{-4}$	4.99 × 10${}^{-4}$	4.03 × 10${}^{-4}$	3.84 × 10${}^{-4}$ (−)
ME	kgNeq	1.13 × 10${}^{-5}$ (+)	5.65 × 10${}^{-6}$	5.69 × 10${}^{-6}$	2.23 × 10${}^{-6}$	1.12 × 10${}^{-6}$ (−)	1.16 × 10${}^{-6}$
MRS	kgCueq	2.04 × 10${}^{-6}$ (+)	1.30 × 10${}^{-6}$	1.16 × 10${}^{-6}$	2.02 × 10${}^{-6}$	1.29 × 10${}^{-6}$	1.15 × 10${}^{-6}$ (−)
OHH	kgNO${}_{x}$eq	1.73 × 10${}^{-5}$ (+)	9.16 × 10${}^{-6}$	9.67 × 10${}^{-6}$	1.65 × 10${}^{-5}$	8.76 × 10${}^{-6}$ (−)	9.27 × 10${}^{-6}$
OTE	kgNO${}_{x}$eq	1.83 × 10${}^{-5}$ (+)	9.69 × 10${}^{-6}$	1.07 × 10${}^{-5}$	1.74 × 10${}^{-5}$	9.26 × 10${}^{-6}$ (−)	1.03 × 10${}^{-5}$
SOD	kgCFC11eq	5.64 × 10${}^{-8}$	2.83 × 10${}^{-8}$	2.14 × 10${}^{-6}$ (+)	5.17 × 10${}^{-8}$	2.59 × 10${}^{-8}$ (−)	2.14 × 10${}^{-6}$
TA	kgSO${}_{2}$eq	3.32 × 10${}^{-5}$ (+)	1.81 × 10${}^{-5}$	1.78 × 10${}^{-5}$	3.18 × 10${}^{-5}$	1.74 × 10${}^{-5}$	1.72 × 10${}^{-5}$ (−)
TE	kg1,4-DCB	1.86 × 10${}^{-2}$ (+)	1.68 × 10${}^{-2}$	1.67 × 10${}^{-2}$	1.69 × 10${}^{-2}$	1.60 × 10${}^{-2}$	1.59 × 10${}^{-2}$ (−)
W	m${}^{3}$	3.11 × 10${}^{-2}$ (+)	1.73 × 10${}^{-2}$	1.69 × 10${}^{-2}$	3.08 × 10${}^{-2}$	1.72 × 10${}^{-2}$	1.67 × 10${}^{-2}$ (−)

**Table 17 materials-15-02986-t017:** Result of the economic assessment of the repair business of producer/remanufacturer for product made from CFRP-ff based on various ranges of material costs.

Scenarios	Cost of CFRP-ff (EUR/kg)	NPV (EUR)	Discounted Payback Period (Years)
S1	51.8	+168,302	5
S2	55.4	+167,629	5
S3	59	+166,957	5

**Table 18 materials-15-02986-t018:** Result of the economic assessment of the repair business of producer/remanufacturer for product made from CFRP-bio based on various ranges of material costs.

Scenarios	Cost of CFRP-bio (EUR/kg)	NPV (EUR)	Discounted Payback Period (Years)
S1	55.4	+167,629	5
S2	59	+166,957	5
S3	62.6	+166,284	5

## Data Availability

The used primary data directly represent laboratory test performed by Politecnico di Milano. Due to confidentiality reasons, these data are and will not be publicly available.

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
