# Peer review of "Environmental and Economic Assessment of Repairable Carbon-Fiber-Reinforced Polymers in Circular Economy Perspective"

_materials, 2022, doi:10.3390/ma15092986_

Round 1
Reviewer 1 Report
This paper deals with the topic on environmental and economical assessment of repairable CFRP. The authors have done a lot of analysis and calculation work. However, further improvements should be made based on the following comments. In addition, the innovation of present research work should also be further highlighted.
*Abstract:
The writing of the abstract needs a high degree of conciseness, only including the research significance, materials and methods, research results and main conclusions.
* Introduction
- As mentioned, the amount of CFRP composite materials is increasing year by year, compared with steel and other metal materials. However, the present summary ignores the important application fields of CFRP closely related to environmental impact. In recent years, the application of CFRP materials in civil engineering, bridge engineering, oilfield engineering and marine engineering has gradually increased, such as bridge cable, prestressed and non-prestressed reinforcers, sucker rods, etc. In the above fields, because CFRP has unique excellent performances, replacing the traditional steel material can significantly reduce the environmental pollution caused by steel materials in the production process. It is suggested that the authors add some relevant summary according to the comments above, and further analyze the effect of CFRP on the environment in different application fields, for example, Composite Structures, 2022. 281: 115060. Composite Structures 2022, 279, 114697. Journal of Materials Research and Technology, 2021, 14:2812-2831.
- Part 1.2, it is suggested that the authors briefly give the main objectives of the current research work. Furthermore, the key problems to be solved further should also be given.
- At present, a lot of research work has been carried out on LCA and LCC. Therefore, the innovation of this research work should be further highlighted. In addition, the summary of others’ research work should be highly concise and in-depth analysis. Furthermore, the unsolved problems should be paid attention to highlight the contribution and significance of this work.
* Materials and Methods
- The research objectives in part 2.1.1 and introduction are duplicate, so it is suggested to delete them.
- After the product is produced, it will be transported to different fields for application. Therefore, the transportation part should include the above part in addition to the transportation of CFRP waste.
- A key problem is that the authors use a lot of space to write the materials and methods by summarizing the research work of others. However, materials and methods are the main research work of the current research, so the summary of other research work and some basic definitions etc. should be written in the introduction. A large number of contents are put together, which is difficult for readers to have a clear understanding. At the same time, it is difficult to emphasize the main contribution of the present work.
*Results and discussion
- Figure 3 shows some evaluation results of LCA, which is suggested to be transformed into table form.
- From the Figure 4 to Figure 7, a large number of indicators and parameters related to the environmental impact are presented. Which parameters have greater impact on the environment? Some quantitative analysis and summary results can be given.
* Conclusions
Conclusions should be rewritten to include only key results and findings.
Reviewer 2 Report
Dear Authors,
I have read manuscript titled: “Environmental and Economical assessment of reparable Carbon Fibre Reinforced Polymers in Circular Economy perspective” with great attention.
In my opinion, the article has a good scientific level is well structured, contains detailed literature review and can be published in current state, as an original and valuable work.
In the presented study the authors perform life cycle assessment and life cycle costing study to obtain a broad assessment on carbon fiber reinforced polymers (CFRP). The study contributes to the development of Circular Economics by emphasizing the importance of simultaneous repairing, recycling and use of bio-based materials throughout production of CFRP. In the pool of the studies on sustainability of CFRPs the presented study distinguishes with attempts to provide a complete assessment of CFRPs throughout applied life cycle assessment and life cycle costing approaches. Moreover, in the scope of studies on the composite materials it outlines the importance of not only recycling but also need on repairing products to reduce their environmental impact.
The paper is well written and structured, thus easy to read and comprehend. The statements of the authors are supported with the adequate and relevant references. The analysis is performed in depth and shows the scientific soundness of the study. The scenarios of the analysis together with the material production processes are well presented in the Figs. 1 and 2. The input data used for the analysis is qualitative tabulated and the outcome is clearly presented. The performed detail analysis is followed by the competent discussion showing the expertise of the authors in the presented topic. The conclusions clearly respond to the objectives of the study presented in the section 1.2, reinforce main idea of the authors, and summarize the study as whole.
Referring to the aforementioned I suggests to publish the article in the present form.
Reviewer 3 Report
Dear Author's
This paper focused on the new types of composites, that can be repaired and recycled. The paper aims to outline the importance of needed efforts on repairing composite products to reduce their environmental impact and resources depletion and lower costs.
The topic is very actual, and interesting from the Environmental and Economic perspective. After a careful assessment, I do think the paper can be published after one minor revision.
- Change the Figure 3 to tables, and insert the units in the correct format as per the guidelines.
Round 2
Reviewer 1 Report
The paper is acceptable, and the authors have made the comprehensive modifications.